# A Novel Approach for the Synthesis of Responsive Core–Shell Nanogels with a Poly(N-Isopropylacrylamide) Core and a Controlled Polyamine Shell

**DOI:** 10.3390/polym16182584

**Published:** 2024-09-13

**Authors:** Anna Harsányi, Attila Kardos, Pinchu Xavier, Richard A. Campbell, Imre Varga

**Affiliations:** 1Institute of Chemistry, Eötvös Loránd University, Pázmány P. s. 1/A, 1117 Budapest, Hungary; 2Department of Chemistry, University J. Selyeho, 945 01 Komárno, Slovakia; 3Division of Pharmacy and Optometry, Faculty of Biology, Medicine and Health, University of Manchester, Oxford Road, Manchester M13 9PT, UK

**Keywords:** responsive nanogels, core–shell nanogels, cationic shell, polyethyleneimine shell, chemical coupling

## Abstract

Microgel particles can play a key role, e.g., in drug delivery systems, tissue engineering, advanced (bio)sensors or (bio)catalysis. Amine-functionalized microgels are particularly interesting in many applications since they can provide pH responsiveness, chemical functionalities for, e.g., bioconjugation, unique binding characteristics for pollutants and interactions with cell surfaces. Since the incorporation of amine functionalities in controlled amounts with predefined architectures is still a challenge, here, we present a novel method for the synthesis of responsive core–shell nanogels (*d_h_* < 100 nm) with a poly(*N*-isopropylacrylamide) (pNIPAm) core and a polyamine shell. To achieve this goal, a surface-functionalized pNIPAm nanogel was first prepared in a semi-batch precipitation polymerization reaction. Surface functionalization was achieved by adding acrylic acid to the reaction mixture in the final stage of the precipitation polymerization. Under these conditions, the carboxyl functionalities were confined to the outer shell of the nanogel particles, preserving the core’s temperature-responsive behavior and providing reactive functionalities on the nanogel surface. The polyamine shell was prepared by the chemical coupling of polyethyleneimine to the nanogel’s carboxyl functionalities using a water-soluble carbodiimide (EDC) to facilitate the coupling reaction. The efficiency of the coupling was assessed by varying the EDC concentration and reaction temperature. The molecular weight of PEI was also varied in a wide range (*M_w_* = 0.6 to 750 kDa), and we found that it had a profound effect on how many polyamine repeat units could be immobilized in the nanogel shell. The swelling and the electrophoretic mobility of the prepared core–shell nanogels were also studied as a function of pH and temperature, demonstrating the successful formation of the polyamine shell on the nanogel core and its effect on the nanogel characteristics. This study provides a general framework for the controlled synthesis of core–shell nanogels with tunable surface properties, which can be applied in many potential applications.

## 1. Introduction

Responsive microgel (*d* ~ 100 nm–2–3 μm) and nanogel (*d* < 100 nm) particles have attracted significant interest in the last few decades due to their versatile applications in fields such as sensing, catalysis, drug delivery and biotechnology [1,2,3,4,5,6]. Many of these applications require not only the incorporation of well-defined chemical functionalities into the gel particles but the localization of these functionalities in various volume elements (e.g., in the core or in the shell) of the particles. As outlined below, one particularly challenging task is to prepare core–shell nanogels with a high-charge-density polyelectrolyte shell. Another challenge in the field is the controlled incorporation of amine functionalities into the microgels in a significant amount. In this work, we present a novel approach that allows for the formation of robust and controlled polyamine shells around a microgel core.

Responsive micro- and nanogels are formed by a water-swollen polymer network that has the ability to undergo a substantial non-linear but reversible volume change due to various external stimuli, such as a change in temperature, pH or ionic strength [7,8]. Among the various stimuli-responsive polymers, poly(*N*-isopropylacrylamide) (pNIPAm) has emerged as one of the most widely studied and utilized materials [9]. pNIPAm has a lower critical solution temperature (LCST) at ~32 °C, which is close to physiological temperature. Below this temperature, the polymer network is hydrated and swollen, but as the LCST is approached, the pNIPAm chains lose their water solubility and the polymer network collapses, resulting in a volume change of more than an order of magnitude in the gel beads. This is often referred to as the volume phase transition (VPT) of the gel particle. Since several monomers can be copolymerized with *N*-isopropylacrilamide, the physical and chemical characteristics of pNIPAm-based microgel particles can be easily tuned. Thus, e.g., the volume phase transition temperature (VPTT), the size, the charge, the desired external stimuli and the chemical functionalities can be adjusted to the desired application [7,9].

Core–shell microgels prepared with a responsive core and a responsive shell can offer significant advantages over simple pNIPAm-based copolymer microgels. Such a microgel architecture offers more sophisticated and tunable responses to external stimuli, allowing for the independent optimization of the core and shell characteristics [10,11]. The shell may act as a protective barrier, controlling access to the core and ensuring that the core’s response is triggered only under precise conditions [12,13]. This can be particularly advantageous in applications such as targeted drug delivery, where the shell can prevent the premature release and degradation of the drug [14]. The shell can also provide improved colloidal stability and enhance the microgel’s performance in complex environments [15,16,17]. Additionally, specific functionalities can also be localized either in the core or in the shell to facilitate the localized modification of the core–shell microgel in post-synthetic reactions [18,19].

The first double-responsive core–shell microgel particles were synthetized by Jones and Lyon [20], who prepared core–shell microgels by introducing acrylic acid either into the core or into the shell, thus creating spatially separated regions within the microgels with different volume phase transition temperatures (VPTTs). Later, Berndt et al. used two different acrylamide monomers (NIPAm and *N*-isopropylmethacrylamide) to prepare core–shell microgels with different VPTTs in the core and in the shell [21]. As was pointed out by Lyon [22], the different transition temperatures of the core and the shell imply the development of stresses at the interface of the two different polymer domains, which has an impact on the swelling of the core–shell microgels. Indeed, these effects have been observed both for core–shell [23,24,25] and for hollow “shell–shell” [26] microgels. It should also be noted that hollow microgel capsules are typically prepared through the synthesis of core–shell microgels containing a sacrificial core [26,27].

Responsive microgel particles are typically prepared by free radical precipitation polymerization. Precipitation polymerization method was first introduced by Pelton et al. [28]. The main advantage of this method is that it can be performed relatively easily in an aqueous medium at relatively low temperatures (60–80 °C) using a wide range of comonomers to introduce specific functionalities. Core–shell microgels are also prepared by precipitation polymerization. In this case, the core particle is prepared first, and then it is used as a seed in a second polymerization step to form the core–shell microgel [20]. The main limitation of this approach is that in a precipitation polymerization reaction, the growing polymer chain must become insoluble in water at the reaction temperature to facilitate the chain collapse and the aggregation of the collapsed growing chain with the seed particle. This means that the concentration of hydrophilic monomers is strongly limited in a precipitation polymerization. The upper concentration limit of a specific monomer is determined by its hydrophilicity. Thus, e.g., acrylic acid can be readily copolymerized at 15–20% into pNIPAm microgels [29], while the incorporation of primary amine monomers is typically limited to a few percent [30,31,32].

As an alternative to the seeded precipitation polymerization technique, Hoare et al. used a semi-batch method to prepare anionic core–shell microgels [33]. They fed an anionic comonomer (methacrylic acid) continuously into the reaction mixture during a precipitation polymerization. They found that, depending on the feeding rate, the anionic monomer could accumulate in the microgel core or in the microgel shell, or it could become uniformly distributed within the microgel particle. Antoniuk et al. [19] used a similar semi-batch precipitation polymerization approach to prepare a core–shell microgel with a pNIPAm core and a practically pure acrylic acid shell. However, instead of the continuous feeding of the anionic comonomer, they only added the acrylic acid to the reaction mixture in the last stage of the core polymerization (the monomer conversion had already exceeded 90%), resulting in the formation of a well-defined acrylic acid shell on the pNIPAm core.

Thermo-responsive core–shell nanogels with primary amine functionalities are highly desirable nanomaterials in several medical applications, e.g., as nanocarriers. Unfortunately, as noted above, the copolymerization of amine monomers into a pNIPAm polymer network in a traditional precipitation polymerization is rather limited due to the highly hydrophilic nature of these monomers [30]. Nonetheless, the graft copolymerization of NIPAm with polyamines could overcome this problem in some cases [34]. In this approach, NIPAm is polymerized through precipitation polymerization in the presence of either polyethyleneimine or chitosan, leading to the formation of temperature-responsive core–shell microgels with polyamines predominantly localized in the shell. Unfortunately, this method produces core–shell microgels only with a very thin polyamine layer and provides very limited opportunities to control the particle characteristics.

The main objective of this work was to develop a simple yet robust method for the synthesis of pNIPAm nanogel particles with a controlled-thickness high-charge-density cationic shell. To achieve this goal, our strategy was to prepare pNIPAm nanogel particles functionalized with carboxylic groups in their surface layer and then use the incorporated carboxylic groups as anchor points to graft primary amine-bearing polycation chains to the nanogel beads (Figure 1). To ensure the simplicity of our approach, the surface-functionalized pNIPAm nanogel beads were prepared in a single-pot reaction, and these core particles were further used for coupling the investigated polyamines. To facilitate the crosslinking between the primary amine groups of the polyamines and the carboxyl groups of the microgel core, we used a water-soluble carbodiimide (*N*-(3-Dimethylaminopropyl)-*N*′-ethylcarbodiimide hydrochloride, EDC). To control the thickness of the polyamine shell on the pNIPAm core, hyperbranched polyethyleneimines of different molecular weights were utilized. It should be emphasized that the coupling could be performed in an aqueous medium at room temperature, thus resulting in an uncomplicated experimental protocol that can be easily replicated with basic laboratory tools.

## 2. Materials and Methods

### 2.1. Materials

*N*-isopropylacrylamide (NIPAm, 99%) was obtained from VWR International Kft., Debrecen, Hungary and was recrystallized from n-hexane. *N*,*N*’-methylenebis (acrylamide) (BA, 99%) was purchased from VWR International Kft., Debrecen, Hungary and was recrystallized from methanol. Acrylic acid (AAc, 99%+) was obtained from Merck Kft., Budapest, Hungary and was purified twice by vacuum distillation prior to usage. Sodium dodecyl sulfate (SDS, 99%+) was received from Merck Kft., Budapest, Hungary and was purified by recrystallization from absolute ethanol. Ammonium persulfate (APS, ACS Grade), *N*-(3-Dimethylaminopropyl)-*N*′-ethylcarbodiimide hydrochloride (EDC), 2-(*N*-morpholino)ethanesulfonic acid hydrate (MES) and copper(II) sulfate were obtained from Merck Kft., Budapest, Hungary and they were used as received. Hyperbranched poly(ethyleneimine)s (PEI) of different molecular weights (*M_w_* = 0.6, 2, 10, 750 kDa) were purchased either from VWR International Kft., Debrecen, Hungary or Merck Kft., Budapest, Hungary and were used as received. Rectapur-activated carbon was purchased from VWR International Kft., Debrecen, Hungary, and it was washed with MilliQ water. All solutions were prepared in ultrapure MilliQ water (total organic content ≤ 5 ppb, resistivity ≥ 18 MΩcm).

### 2.2. Synthesis of the Carboxyl-Functionalized pNIPAm Nanogel Core

Carboxyl-functionalized core–shell nanogel particles were synthesized by free radical precipitation polymerization in an aqueous solution using a modified version of the method developed by McPhee et al. [35]. In practice, 450 mL of MQ water was placed in a double-wall glass reactor and purged with argon at 80 °C for one hour. After an hour, 92 mL of the purged water was removed from the reactor and stored under argon pressure. NIPAm (4.205 g) and BA (0.191 g) were measured into a 50 mL Pyrex flask and flushed with argon for 10 min; then, the monomers were dissolved by the addition of 32 mL of oxygen-free water. A total of 0.943 g of NIPAm, 0.048 g of BA and 0.069 g of AAc were measured into a 20 mL glass vial. The vial was flushed with argon, and then 12 mL of oxygen-free water was added and stored under argon until used. A total of 0.969 g of SDS was added into a 20 mL glass vial. The vial was flushed with argon, and then 12 mL of oxygen-free water was used to dissolve the surfactant. A total of 0.548 g of APS was measured into another 20 mL vial and flushed with argon, and then 10 mL of oxygen-free water was used to dissolve the material. Once all the solutions were prepared, 30 mL of the NIPAm/BA stock solution and 10 mL of the SDS stock solution were added to the reaction vessel and purged with argon for an additional 15 min. Finally, to start the polymerization reaction, 2 mL of the APS stock solution was quickly injected into the reaction mixture. After 20 min, 10 mL of the AAc-containing stock solution was also injected into the reaction mixture to facilitate the formation of the carboxyl-functionalized shell. The reaction proceeded for four hours, and it was stopped by rapidly cooling the mixture and bubbling oxygen. To follow the monomer conversion, samples were regularly taken and were analyzed by HPLC-UV.

The synthetized nanogel dispersion was purified by a Vivaflow-200 cross-flow filtration cassette (PES, molecular weight cut-off (MWCO) of 100 kDa) to remove the oligomer by-products and some of the surfactant. Prior to the first filtration step, the pH of the dispersion was set to seven. Then, the dispersion was diluted with Milli-Q water to 1 L and concentrated to 200 mL using the filtration device. After three filtration cycles, 20 g of activated carbon was added to the 200 mL of nanogel dispersion and gently stirred for four days. Then, the activated carbon was removed by filtration, and a new batch of the adsorbent was added. After an additional four days of stirring, the activated carbon was removed again by filtration. The purity of the nanogel dispersion was checked by taking a small aliquot (3 mL) and separating the supernatant by ultrafiltration using an AmiconUltra 4 mL centrifugal filter (obtained from Merck Kft., Budapest, Hungary) (MWCO = 3 kDa, regenerated cellulose). Then, the surface tension of the filtrate was measured by the pendant drop method. Since the measured surface tension was above 70 mN/m, we concluded that the purified microgel dispersion could contain SDS only in trace amounts. Finally, the nanogel dispersion was freeze-dried and stored at room temperature in a desiccator before further use.

### 2.3. Grafting the Polyamine Shell to the pNIPAm Core Particles

First, a 1.0 weight% (w%) nanogel stock solution (corresponding to a 1.6 mM nominal AAc concentration) was prepared by dissolving a calculated amount of the lyophilized nanogel in MilliQ water. Stock solutions of 1.0 w% were also prepared from all the different-molecular-weight PEIs. In this case, 2.0 w% stock solutions were prepared first, and then the pH of these solutions was adjusted to 5.5 using HCl. Then, they were diluted with MilliQ water to obtain the 1 w% stock solutions. Finally, a 500 mM MES buffer at pH = 5.5 was also prepared. All the stock solutions were filtered through a 0.8 μm cellulose acetate membrane filter prior to use.

All the polyamine coupling was conducted in 10 mL sample volumes. First, 5 mL of the nanogel stock solution was pipetted into a 20 mL glass vial. Then, 1 mL of the MES stock solution and 1 mL of a PEI were added. Before initiating the binding reaction, the prepared mixture was allowed to equilibrate to a temperature of either 25 °C or 4 °C for 30 min. Finally, a calculated amount of EDC was dissolved in 3 mL of temperature-equilibrated water and added to the reaction mixture to obtain the 10 mL final volume. The final concentration of the nanogel was always 0.5 w% (0.8 mM nominal carboxyl concentration), while the PEI concentration was 1000 ppm (0.1 w%) in the reaction mixture. The EDC concentration varied from a stoichiometric amount to an excess of fifteen times (1, 2, 5, 7.5, 10 and 15 times) compared to the carboxyl concentration in the case of the 0.6 kDa PEI coupling. In all other cases, an EDC excess of ten times was used. The reaction was allowed to proceed for two days in each case.

Three of the four prepared core–shell nanogels were purified by ultrafiltration using Amicon Ultra 15 mL centrifugal filters. The ultrafiltration membranes had an MWCO of 30 kDa in the case of 0.6 and 2 kDa PEI coupling and 100 kDa in the case 10 kDa PEI coupling. The samples were filtered three times, and in all cases, the PEI concentration in the final filtrate was under 1 ppm. When the 750 kDa PEI was coupled to the nanogel core, purification was carried out by ultracentrifugation using an Optima XPN-100 type ultracentrifuge (Beckman Coulter, Indianapolis, IN, USA) equipped with a Type 90 Ti rotor and OptiSeal polycarbonate centrifuge tubes. The centrifugation was conducted at 25 °C with a speed of 65,000 rpm (362,000× *g*). The centrifuged core–shell nanogels were redispersed in MilliQ water, and the centrifugation was repeated two more times.

### 2.4. Dynamic Light Scattering

The hydrodynamic size of the prepared nanogels was determined by dynamic light scattering (DLS). The measurements were performed using a Brookhaven Instruments (Nashua, NH, USA) light scattering device containing a BI-200SM goniometer and a BI-9000AT digital autocorrelator. A Coherent Genesis MX488-1000 STM (Saxonburg, PA, USA) laser was used as a light source. The laser emitted vertically polarized light at a wavelength of 488 nm. The time axis of the autocorrelator was set logarithmically to span the required correlation time range. The autocorrelation functions were measured at a scattering angle of 90° with a 100 μm pinhole size. The obtained autocorrelation functions were analyzed by the second-order cumulant and the CONTIN methods. Since the CONTIN evaluation always showed narrow monomodal size distributions, we used the effective diameter (*d_h_*) determined by the second-order cumulant method to characterize the hydrodynamic size of our samples. Each data point determined by dynamic light scattering was the average of at least ten measurements. The standard error of the datapoints was typically a few nanometers, which is commensurate with the size of the symbols in the presented graphs.

### 2.5. Electrophoretic Mobility Measurements

A Malvern Zetasizer Nano ZS (Malvern, Worcestershire, UK) device was used to measure the electrophoretic mobility (*μ_E_*) of the prepared core–shell nanogels. The instrument used a combination of laser Doppler velocimetry and phase analysis light scattering (PALS) in a technique called M3-PALS. Before the measurements, the instrument was always tested with a Malvern Zeta refence sample. All the measurements were performed at 25 °C. At least five parallel measurements were made for each sample, and the relative standard error of the mean electrophoretic mobility was approximately 5–10%.

### 2.6. PEI Binding Measurements

The amount of the nanogel-bound PEI was determined by a method described previously by Wen et al. [36]. Briefly, 10 mM CuSO4 was added to a PEI filtrate of an unknown concentration at a 1:1 volume ratio, and the absorbance of the solution was determined at 620 nm. The concentration of the PEI solution was determined using a calibration curve measured at a constant 5 mM CuSO4 concentration. An individual calibration curve was determined for each of the different-molecular-weight PEI solutions. The amount of the nanogel-bound PEI (*c_MG-PEI_*) was determined as the percentage of the analytical concentration of PEI added to the reaction mixture (*c*_*PEI,*0_):
$$c_{MG-PEI}=\frac{c_{PEI,0}-c_{PEI,free}}{c_{PEI,0}}100\%$$
where *c_PEI,free_* indicates the PEI concentration in the filtrate. A Perkin Elmer Lambda 2 UV/Vis spectrophotometer (Shelton, CT, USA) was used to record the spectra of the PEI/CuSO_4_ mixtures in the 400–800 nm wavelength range in a glass cuvette with a path length of 1.00 cm.

## 3. Results and Discussion

To facilitate the addition of a robust polycationic shell to a pNIPAm nanogel core, a surface-functionalized poly(*N*-isopropylacrylamide) (pNIPAm) nanogel was first synthesized. Carboxyl functionalities were incorporated into the outer shell of the synthesized core particles to enable polyamine coupling. In this context, it is important to note that we aimed at confining the carboxyl functionalities to the surface layer of the core particles to prevent the potential coupling of polyamine chains within the interior of the core particles, which could compromise the temperature-responsiveness of the pNIPAm core. To achieve this goal, a single-pot two-step precipitation polymerization was used to prepare the core particles. Our approach relied on previous findings indicating that once a nanogel bead is formed at the initial stage of particle synthesis, the composition of the polymer chains growing on the surface of these gel beads is determined by the monomer composition of the reaction mixture [19,33,37]. Thus, we first initiated the precipitation polymerization of the pNIPAm nanogels in a reaction mixture that contained only *N*-isopropylacrylamide (NIPAm) and a methylene-bis-acrylamide (BA) crosslinker. After 20 min, when most of the initial monomers were practically consumed (monomer conversion exceeded 90%) and the pNIPAm nanogel beads were already formed, a second batch of monomers was added to the reaction mixture. However, this batch of monomers also contained acrylic acid (AAc) in addition to NIPAm and BA. The molar ratio of the monomers in the initial reaction mixture and in the second batch was 9 to 2, and 10% of the monomers in the second batch was acrylic acid. Assuming that the segment density of the growing collapsed nanogels was practically constant, the acrylic acid was confined in the outmost shell of the gel beads that had a shell thickness of ~7% of the microgel radius.

Another important aspect of the nanogel core synthesis was that since we wanted to keep the size of the swollen core particles below 100 nm, the core synthesis was performed in the presence of a high sodium dodecyl sulfate (SDS) concentration (*c_SDS_* = 7 mM) [38], which had an important impact on the purification of the core particles. In a typical microgel (nanogel) synthesis, the gel particles are purified with repeated (two or three times) centrifugations. Unfortunately, in the present case, this approach could not be used efficiently, since at room temperature, SDS binds cooperatively to pNIPAm microgels with a critical aggregation concentration (*cac*) of ~1 mM [38]. This means that the supernatant would contain SDS only at a 1 mM concentration, and most of the surfactant would remain bound to the centrifuged nanogel beads. Thus, the removal of the surfactant would require a large number of centrifugation/redispersion cycles, which could only be performed by ultracentrifugation due to the small size and highly swollen character of the gel nanobeads. To avoid this tedious procedure, we performed ultrafiltration using a cross-flow cell (Vivaflow, MWCO = 100 kDa) to remove the oligomer byproducts and some of the surfactant from the reaction mixture. Next, the rest of the surfactant was removed by repeated adsorption on activated carbon. To confirm the successful removal of the surfactant, a small aliquot of the purified nanogel dispersion was ultrafiltered by an AmiconUltra centrifugal filter (MWCO = 3 kDa), and the surface tension of the filtrate was determined by the pendant drop method. Since the surface tension of the filtrate remained above 70 mN/m, we concluded that only a trace amount of SDS could have remained in the nanogel dispersion after the applied purification protocol.

Finally, we would like to emphasize that the experimental protocol described in the Material and Methods section for the semi-batch polymerization of the pNIPAm nanogels with carboxyl surface functionalities ensured the full control of the polymerization kinetics; thus, the control and reproducibility of the particle size and functionalization efficiency was possible. At the same time, the synthesis could be performed in any flask equipped with a septum that facilitates the bubbling of the reaction mixture with an inert gas and the injection of the initiator and a second batch of monomers, as long as the temperature is maintained within a few degrees and oxygen is excluded from the reactor. Less-well-defined synthetic conditions may decrease the yield and would decrease the quantitative control and reproducibility of the particle characteristics but could allow for the formation of the desired microgel architecture. 

To confirm the successful incorporation of the acrylic acid into the nanogel beads, we measured the temperature-dependent swelling of the prepared nanogel particles (Figure 1a). The measurements were performed both at pH = 7, where the carboxylic groups were fully dissociated (charged), and at pH = 2, where they were protonated (uncharged) (*pK_a_* = 4.25 [39]). As shown in Figure 1a, below the volume phase transition temperature of pNIPAm (*VPTT_pNIPAm_* = 32 °C), the nanogel particles had a significantly larger hydrodynamic size at the higher pH. This clearly shows the enhanced swelling of the nanogel particles due to the charged carboxylic groups incorporated into the polymer network. Furthermore, the presence of these charged groups also increased the VPTT of the polymer network and provided colloid stability for the collapsed nanogel beads at elevated temperatures at the applied ionic strength. At the same time, at the low pH, where the carboxylic groups were uncharged, the collapsed gel beads lost their colloid stability and aggregated above the VPTT. This implies that the protonated carboxyl groups were localized in the surface layer of the collapsed particles, resulting in the loss of the electric double layer of the collapsed nanogel particles, which provided colloid stability at high pH. As a reference, the temperature dependence of the hydrodynamic size of a non-functionalized pNIPAm microgel prepared in an independent batch precipitation polymerization reaction is also plotted in Figure 1b. As is clearly shown by the figure, in this case, the pH had a minor effect on both the VPTT and the swelling of the microgel particles. This is in sharp contrast with the results we obtained for the carboxyl-functionalized microgel, highlighting the successful incorporation of the acrylic acid functionalities during the semi-batch polymerization. It is also interesting to note that the non-functionalized pNIPAm microgel preserved its colloid stability above its VPTT, even at the low pH. This indicates that in this case, the surface charge of the microgel was determined by the sulfate groups originating from the initiator.

To further characterize the prepared nanogel particles, their electrophoretic mobility and hydrodynamic size were also measured as a function of pH at room temperature (25.0 °C). The results of these measurements are plotted in Figure 2. As shown in the figure, a stepwise increase can be observed in both the hydrodynamic size and the electrophoretic mobility of the nanogels between pH ≈ 3 and pH ≈ 6. Considering that acrylic acid has a *pK_a_* = 4.25 [39], these observations also confirm the incorporation of the acrylic acid monomer into the surface layer of the nanogel beads.

To form the core–shell nanogels with a polyamine shell, we aimed at the chemical coupling of polyamines to the carboxyl functionalities of the nanogel particles. To achieve this goal, we used a water-soluble carbodiimide (*N*-(3-Dimethylaminopropyl)-*N*′-ethylcarbodiimide hydrochloride, EDC) to form a reactive O-acylisourea intermediate ester from the carboxyl groups, which is a well-established method to couple carboxyl groups to primary amines in aqueous media [40]:



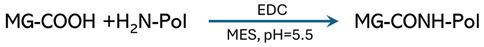



A drawback of EDC coupling in aqueous media is that water molecules can also react with the active ester, thus regenerating the initial carboxyl groups. Furthermore, water can also react directly with EDC [40]. Thus, EDC is typically used in excess during the coupling reaction. To establish an optimized coupling protocol, we used a reaction mixture that contained the nanogel particles at a fixed 0.5 w% concentration, resulting in a 0.8 mM nominal acrylic acid concentration, and a low-molecular-weight (*M_w_* = 0.6 kDa) hyperbranched polyethyleneimine (PEI). The PEI was used at a 1000 ppm (24 mM repeat unit) concentration. Since the hyperbranched PEIs contained the primary, secondary and tertiary amine groups at a 1:2:1 ratio [41], this represents a 6 mM primary amine concentration in the reaction mixture, which corresponded to a primary amine excess of ~8 times compared to the carboxyl concentration. Since the direct hydrolysis of EDC is catalyzed by acids and alkalis, EDC coupling is typically carried out between pH 5 and 7 [42,43]. Here, we set the pH of the reaction mixture to 5.5 using a 50 mM MES buffer. To optimize the EDC excess, the coupling reaction was performed from a stoichiometric EDC concentration to an excess of 15 times. After the reaction was completed, the nanogel particles were purified by repeated ultrafiltration using an Amicon centrifugal filter (MWCO = 30 kDa). To characterize the coupling efficiency, the PEI concentration in the first filtrate and in the purified nanogel suspensions were determined using a spectrophotometric method. The sum of the free and the nanogel-bound PEI concentrations were determined, and they were found to be equal to the initial PEI concentration within the experimental error for each sample, indicating the reliability of the binding measurements. It should be noted that since the rate of the direct hydrolysis of EDC significantly depends on the temperature, EDC coupling is often carried out at a low temperature (4 °C) to suppress hydrolysis. To assess the effect of the temperature on the efficiency of the coupling reaction, we performed the reaction series both at 4 °C and at room temperature (25 °C). The amount of the nanogel-bound PEI is plotted in Figure 3 as a function of the EDC concentration in both cases.

At an EDC excess of less than five times, the low-temperature coupling indeed provided a better coupling efficiency, giving rise to a ~4% excess PEI binding. However, with increasing the EDC concentration, the efficiency of the PEI binding significantly increased regardless of the reaction temperature, and when a high enough EDC concentration was reached, the amount of microgel-bound PEI leveled off at ~20% of the total initial PEI concentration, with only a minor excess in the case of the low-temperature coupling. Thus, we concluded that using an EDC excess of ten times and performing the coupling at room temperature represents the most convenient but efficient coupling protocol. Considering that the average molecular weight of the bound PEI molecules was 0.6 kDa (14 repeat units), it is straightforward to show that the observed 20% PEI coupling means that the average number of carboxyl groups available for a PEI molecule on the nanogel surface was ~2.

To assess the effect of the developing PEI shell on the swelling characteristics of the prepared core–shell nanogels, the hydrodynamic size and the electrophoretic mobility of two selected core–shell nanogels prepared at 25 °C were determined as a function of the solution pH. The first sample was prepared with a stoichiometric amount of EDC, and the second sample was prepared with an EDC excess of ten times, resulting in couplings of ~10 and ~19% of the initial PEI concentration to the nanogels, respectively. The results of the measurements are plotted in Figure 4, and as a reference, the results obtained previously for the bare core particles are also plotted. As shown in the figure, the PEI shell had a profound effect on the characteristics of the nanogel particles. Interestingly, in 10 mM NaOH (pH = 12), both core–shell nanogel particles had a slight negative charge (Figure 4a). This shows that some of the carboxylic groups did not react with the PEI amine groups even at the highest applied EDC concentration; thus, they provided a negative charge to the nanogel particles when the coupled PEI molecules were fully deprotonated (uncharged). However, with decreasing the pH, the coupled PEI molecules became protonated, and the prepared core–shell nanogels first became charge-neutralized and then gained a positive surface charge. This occurred only at pH~6 when a stoichiometric amount of EDC was used in the coupling reaction, but it occurred already at pH~9.5 in the other case. This indicates that a comparable amount of unreacted carboxyl groups and protonated amine groups were present in the core–shell nanogel beads around these pH values, implying the presence of significantly less residual carboxyl groups when EDC was used in a large excess during the coupling reaction.

This interpretation was also confirmed by the swelling curves of the core–shell nanogels. Both samples showed a minimal swelling around the same pH value where their charge reversal occurred (Figure 4b). This minimum was due to the charge compensation of the oppositely charged carboxylic and protonated amine groups, which were present in similar amounts at these pH values in the prepared core–shell nanogels. When the pH was decreased, the positive charges of the PEI shell were in excess; when the pH was increased, then the negative charges of the residual carboxyl groups were in excess. At the same time, these excess charges gave rise to the swelling of the nanogel particles, as can be seen in Figure 4b. However, it should be noted that that while in the case of the stoichiometric EDC coupling, the dominant swelling occurred due to the residual carboxyl groups, in the other case, the carboxyl groups had only a minor effect on the nanogel swelling, which was dominated by the charging up of the PEI shell with decreasing pH.

Since we found that the PEI molecules coupled with the nanogel core through only a few of their amine groups, we hypothesized that by increasing the molecular weight of the coupled PEI molecules, more PEI repeat units could be immobilized on the nanogel surface, resulting in the formation of a denser and/or thicker polyamine shell. To assess this possibility, we carried out a series of coupling reactions using the previously defined optimized coupling protocol (using an EDC excess of ten times at room temperature). While the PEI concentration was kept constant (1000 ppm), the molecular weight of the coupled PEI varied from 0.6 kDa to 750 kDa. The amount of nanogel-bound PEI was determined for each sample, and the results are plotted in Figure 5.

As shown in the figure, the amount of the nanogel-coupled PEI indeed increased with increasing the PEI molecular weight used in the reaction, but as the polymer molecular weight reached 10 kDa, it leveled off at ~65% binding. This represents an increase of more than three times compared to the low-molecular-weight (0.6 kDa) coupling. To investigate how the increased PEI binding affects the characteristics of the core–shell nanogels, the electrophoretic mobilities of the prepared samples were measured as a function of the sample pH. As shown in Figure 6, as the PEI molecular weight increased, the negative charge of the core–shell nanogels diminished at a high pH. This indicates that the amount of residual carboxyl groups was reduced to practically zero with increasing the PEI molecular weight, which can be rationalized in terms of an increase in the polymer coil size that provides high local amine concentrations over the entire nanogel surface, which in turn facilitates the coupling reaction. In addition, with the increase in the PEI molecular weight, the core–shell nanogels exhibited a larger positive electrophoretic mobility value at any pH, indicating the incorporation of more and more amine groups into the PEI shell of the nanogel particles.

Finally, as a representative example, the temperature-dependent swelling of the core–shell nanogels decorated with the 10 kDa PEI molecules is also plotted in Figure 7, both at high and at low pHs. As shown in the figure, the hydrodynamic size of the core–shell nanogel was always larger at a low pH, where the PEI shell was charged. At the same time, the size of the uncharged core–shell nanogels was only moderately (a few nanometers) smaller at any temperature, indicating the formation of a dense PEI layer, where presumably each PEI molecule was coupled to the nanogel core at several points, thus limiting the swelling of the PEI shell. 

The figure also indicates a steep particle size decrease around the VPTT of the pNIPAm network (~32 °C) regardless of the solution pH. This highlights the dominant effect of the particle core on the swelling characteristics of the nanogel particle. At the same time, the hydrophilic PEI shell hindered the full collapse of the particle core, which resulted in a gradual particle size decrease with the further increase in temperature, and this effect was more pronounced at a low pH, where the PEI shell was charged.

## 4. Conclusions

In this study, we developed a robust and adaptable method for synthesizing responsive core–shell nanogels with a poly(*N*-isopropylacrylamide) (pNIPAm) core and a polyamine shell. Our approach effectively confined carboxyl functionalities to the outer shell of the nanogel core, ensuring the preservation of the core’s temperature-responsive properties while allowing for successful polyamine coupling. The optimization of the EDC-mediated coupling reaction, particularly in terms of the reactant concentration and temperature, was critical in achieving efficient and stable polyamine shell formation. The resulting core–shell nanogels exhibited distinctive pH-dependent electrophoretic mobility and swelling characteristics, highlighting the significance of the coupled polyamine’s molecular weight. While we used polyethyleneimines of different molecular weights in this study to demonstrate the efficiency and simplicity of the proposed method, in principle, the described protocol could be used for the coupling of any amine-functionalized, water-soluble macromolecules (e.g., synthetic and natural polyamines, like polylysine, chitosan, polypetides and amine-functionalized RNAs) to the carboxyl functionalities of the pNIPAm core, offering a scalable and customizable approach to produce core–shell nanogels with tailored surface properties. The versatility of this method may open up potential applications in drug delivery, sensing and other fields, where precise control over the nanomaterial’s properties is required. Future work could explore the functionalization of these nanogels with specific targeting ligands or investigate their assembly with other building blocks to create responsive, 3D-hierarchical nano-assemblies.

Finally, we want to highlight that the PEI shell of the –shell microgels prepared in this work may also offer several potential applications., e.g., this shell may serve as a base layer for the formation of polyelectrolyte multilayer nanocapsules with presumably well-defined capsule permeability, controllable mechanical and surface characteristics and temperature responsiveness. The prepared nanocapsules may be used as building blocks in responsive nanogel multilayers or in formulations of injectable gels. Furthermore, amine groups readily participate in reactions like Schiff’s base or amide formation reactions, making the prepared core–shell nanogels ideal for bioconjugation, thus facilitating the development of advanced biosensors, e.g., for medical diagnostics, bioimaging or environmental monitoring. Last but not least, amines can bind very efficiently with heavy metal ions through chelate formation, making these particles potentially useful in removing contaminants from water sources.

## Data Availability

Data are contained within the article.

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
