# Peer review of "A Novel Approach for the Synthesis of Responsive Core–Shell Nanogels with a Poly(N-Isopropylacrylamide) Core and a Controlled Polyamine Shell"

_polymers, 2024, doi:10.3390/polym16182584_

Round 1

Reviewer 1 Report

Comments and Suggestions for Authors

Thank you to the authors for an interesting work. The development of pH and temperature-sensitive nanogels is an intriguing research direction. However, there are a few minor remarks regarding the presented work:

1. The abstract lacks a discussion on the relevance of the research.

2. The article does not include images of the microgels, which complicates the understanding of the work.

3. Figure 1: The graph line exceeds the maximum value on the Y-axis. Additionally, there are no data points for temperatures above 32°C.

4. The abbreviation "MWCO" requires explanation.

5. The work lacks statistical analysis.

6. The country of origin for the materials is not mentioned.

7. I recommend that the authors discuss the potential applications of the obtained results more specifically.

Reviewer 2 Report

Comments and Suggestions for Authors

The paper presents a novel method for synthesizing responsive core-shell nanogels with a poly(N-isopropylacrylamide) (pNIPAm) core and a polyamine shell by functionalizing its surface with carboxyl groups for chemical coupling with polyamines. Using a two-step precipitation polymerization technique and varying the molecular weight of polyethyleneimine (PEI) and the conditions of the reaction (pH, temperature), the article provides a framework for creating core-shell nanogels with tunable surface properties, potentially useful in applications like drug delivery and biotechnology. Overall, the manuscript provides important research insights and offers a novel approach in the field which as the authors state can be tuned for a specific application, the research and analysis of the findings are comprehensive and reasonable, the text is understandable and fluently written; however, several small improvements are needed.

Some improvements needed are stated below:

Comments:

1)     The title seems confusing – especially the words “general approach”. I think this can be narrowed down in a way that would showcase the novelty of your method more effectively.

2)     Infographics only appear on page 6 and the research paper genuinely reads like a book chapter with no illustrations and references, which might lead to the readers losing their attention spans.

3)     SEM images of the nanogels and their cross section with the shell on the images can be inserted.

4)     The actual topic of the research only comes up on Page 3, with previous pages detailing the history of developments in the designated field. While it is important to provide context and highlight the importance of the research, the theoretical and historical background can be shortened in a way to both convey the same effect and present the actual research question much earlier without the audience having to look through pages to find the point of this paper.

5)     Moreover, comparison characterizations proving successful synthesis and grafting and enhanced electrophoretic mobilities are not shown. While pH and temperature variations and the dependence of the nanogels performance on these variables are shown, actual contrasts with the unmodified starting materials are nowhere to be found.

6)     While the synthesis procedure was laid down in detail, the actual reasoning for some of the additional steps was not provided and the synthesis itself was not discussed at length. Additionally, the synthesis does not seem simple nor accessible at first glance, as the title and the paper suggest, so acknowledgment of that and a reconsideration of the title may be needed.

Comments on the Quality of English Language

No serious issues were found
